# Universal and Efficient Detection of Adversarial Data through Nonuniform Impact on Network Layers

**Furkan Mumcu**                                                       *furkan@usf.edu*
*Department of Electrical Engineering*
*University of South Florida*

**Yasin Yilmaz**                                                       *yasiny@usf.edu*
*Department of Electrical Engineering*
*University of South Florida*

**Reviewed on OpenReview:** *https: // openreview. net/ forum? id= OCY5APFnFI*

## Abstract

Deep Neural Networks (DNNs) are notoriously vulnerable to adversarial input designs with limited noise budgets. While numerous successful attacks with subtle modifications to original input have been proposed, defense techniques against these attacks are relatively understudied. Existing defense approaches either focus on improving DNN robustness by negating the effects of perturbations or use a secondary model to detect adversarial data. Although equally important, the attack detection approach, which is studied in this work, provides a more practical defense compared to the robustness approach. We show that the existing detection methods are either ineffective against the state-of-the-art attack techniques or computationally inefficient for real-time processing. We propose a novel universal and efficient method to detect adversarial examples by analyzing the varying degrees of impact of attacks on different DNN layers. Our method trains a lightweight regression model that predicts deeper-layer features from early-layer features, and uses the prediction error to detect adversarial samples. Through theoretical arguments and extensive experiments, we demonstrate that our detection method is highly effective, computationally efficient for real-time processing, compatible with any DNN architecture, and applicable across different domains, such as image, video, and audio.

## 1 Introduction

Deep neural networks (DNNs) are known to be vulnerable to subtle and manipulative noise for input data instances that can be designed by adversaries to cause erroneous outputs. Notably, Goodfellow et al. (2014) proposed a simple and effective method, called the Fast Gradient Sign Method (FGSM), to craft such adversarial instances by adding or subtracting a small perturbation to each input dimension based on the sign of the gradient. After FGSM, various adversarial sample generation methods were demonstrated across different domains. However, compared to the vast diversity among attack techniques, there are not enough studies that effectively and efficiently detect the increasing number of attacks in various domains.

There are two main defense strategies. The more ambitious one aims to mitigate the effects of attacks (e.g., correctly classifying adversarial images) by developing robust DNNs that are increasingly less vulnerable to adversarial data. As opposed to this idealistic objective, the more practical second strategy aims to detect and discard the adversarial data. This binary classification approach (natural vs. adversarial), which we use in this work, naturally provides a more tractable defense strategy than trying to solve the original DNN problem with adversarial data.

One of the oldest and most effective defense techniques is adversarial training for both the robustness Goodfellow et al. (2014) and detection Grosse et al. (2017), in which the DNN is simply retrained using

also the known adversarial instances. Although effective against known attacks, adversarial training suffers from high computational cost and unseen attacks. Similarly, other detection methods are typically only effective against some attacks. For instance, Metzen et al. (2017), in a supervised learning setup, trains a binary classifier for attack detection, which performs well on the attacks seen in training, but fails for the unseen attacks. A recently proposed detector, EPS-AD Zhang et al. (2023), can detect unseen attacks by training only on natural data in a semi-supervised anomaly detection setup at the expense of significant computational cost. Another recent detection method, VLAD Mumcu & Yilmaz (2024c), can also detect a wide range of attacks by training a secondary model only on natural data, however it is vulnerable to transferable attacks that also affect its secondary baseline model.

The existing adversarial data detectors in the literature are proposed for a particular application and do not naturally extend to other applications as they are based on domain-specific DNNs, e.g., adversarial image detection using convolutional neural networks Metzen et al. (2017) or diffusion models Zhang et al. (2023). Motivated by the lack of a computationally efficient detector which can accurately detect a wide range of attacks in real-time, we propose a *universal and efficient* detector, called Layer Regression (LR), that works together with DNNs in a variety of applications, including image recognition, video action recognition, and speech recognition. Our contributions can be summarized as follows:

- We propose the first universal and efficient adversarial data detection method, which takes advantage of the nonuniform impacts of adversarial samples on different DNN layers.

- We conduct extensive experiments for image recognition, and show that LR significantly outperforms existing efficient methods. Universality of LR is shown with its superior performance in detecting action recognition attacks and speech recognition attacks.

- In addition to its high performance across a wide range of domains, models, and attacks, LR is also the most lightweight defense method. It is orders of magnitude faster than the existing defenses, making it ideal for real-time attack detection and resource-constrained systems.

## 2 Related Work

### 2.1 Adversarial Attacks

The robustness of deep neural networks and their vulnerability to adversarial examples have been investigated since the introduction of FGSM (Goodfellow et al., 2014). Numerous adversarial attacks have been proposed to generate effective adversarial examples in recent years (Madry et al., 2017; Croce & Hein, 2020b; Kurakin et al., 2018; Chen et al., 2017; Ilyas et al., 2018; Mumcu & Yilmaz, 2024a; Wang & He, 2021; Fang et al., 2024; Gao et al., 2020). There are two main adversarial attack settings, namely white-box and black-box. While it is assumed that the attacker has access to the target model in the white-box setting, in the black-box setting, the attacker does not have any prior information about the target model.

White box attacks, including FGSM (Goodfellow et al., 2014), PGD (Madry et al., 2017), APGD (Croce & Hein, 2020b), generate adversarial examples by maximizing the target model's loss function and they are usually referred to as gradient-based attacks. BIM (Kurakin et al., 2018) tries to improve gradient-based attack by applying perturbations iteratively. Transferability based black-box attacks, which were introduced in Papernot et al. (2017), among one of the most common black-box approaches. The idea is to use an attack on a substitute model for generating adversarial examples for unknown target models, utilizing the transferability of adversarial examples to different DNNs. While adversarial examples generated by these attacks are most effective when the substitute model exactly matches the target model as in a white-box attack setting, their success is shown to be transferable even when there is significant architectural differences between the substitute and target models. Wang & He (2021) introduced VMI and VNI to further extend iterative gradient-based attacks and try to achieve high transferability by considering the gradient variance of the previous iterations. PIF (Gao et al., 2020) uses patch-wise iterations to achieve transferability. ANDA (Fang et al., 2024) aims to achieve strong transferability by avoiding the overfitting of adversarial examples to the substitute model. In addition, there are approaches that combine multiple attacks, such as AutoAttack (Croce & Hein, 2020a), to test the robustness of models against a diverse set of adversarial perturbations.

## 2.2 Adversarial Defenses

Attempting to make changes on the input data for removing the effects of perturbations from adversarial examples is the most common defense strategy. JPEG compression is studied in several works (Cucu et al., 2023; Aydemir et al., 2018; Das et al., 2018), and it has been shown that compressing and decompressing helps to remove adversarial effects from input images. Xie et al. (2017) uses random resizing and padding on the inputs to eliminate the adversarial effects. Several denoising methods (Liao et al., 2018; Xiong et al., 2022; Salman et al., 2020) were proposed to remove adversarial perturbations from the inputs. Mustafa et al. (2019) uses wavelet denoising and image super resolution as pre-processing steps to create a defense pipeline against adversarial attacks. Prakash et al. (2018) tries to eliminate adversarial effects by redistributing the pixel values via a process called pixel deflection. Adversarial training is another method which is studied to increase robustness of DNNs, however adversarial training often fails to perform well especially under various attack configurations (Bai et al., 2021). Papernot & McDaniel (2018) proposes to increase the robustness of the model by applying k-nearest neighbors (kNN) classification to the feature representations at different layers of a deep neural network. This method also implicitly utilizes the nonuniform impact of adversarial data on network layers, however their deep kNN method is not universally applicable to any deep neural network and not computationally efficient, unlike our proposed LR method. On the other hand, several detection methods were proposed in recent years. Xu (2017b) introduces feature squeezing where they use bit reduction, spatial smoothing, and non-local means denoising to detect adversarial examples. Zhang et al. (2023) tries to detect adversarial samples by computing an Expected Perturbation Score (EPS), which averages a sample's behavior over multiple perturbations generated using a pre-trained diffusion model. Yang et al. (2022) detects adversarial examples by identifying semantic contradictions by using a generator that reconstructs the input based on the network's internal feature representations. Pang et al. (2018) proposed a training-time modification using reverse cross-entropy to enforce separable feature representations for clean and adversarial inputs, which improves detection performance but requires modifying the training process. Similarly, Tian et al. (2018) introduced a detection method based on prediction consistency under image transformations, leveraging the instability of adversarial samples under rotation or translation, whereas our method operates on internal feature inconsistencies without needing input perturbations or model retraining. A recent defense strategy which utilizes a baseline model, e.g., Vision Language Models (VLMs), with the assumption that the output of target model and baseline model are close to each other for clean input, but are far away from each other for adversarital input. A recent example is demonstrated by Mumcu & Yilmaz (2024c) where they used CLIP (Radford et al., 2021) to detect adversarial video examples.

## 3   Detecting Adversarial Examples

Consider a Deep Neural Network (DNN) model $g(\cdot)$ that takes an input $x$ and predicts the target variable $y$ with $g(x)$. As discussed in Section 2.2, there are three main defense strategies against adversarial attacks: adversarial training, modifying input, and detecting adversarial samples by monitoring changes in output with respect to a baseline. While the former two focuses on the changes in the input ($x$ vs. $x^{adv}$), the latter utilizes the changes in the output ($g(x)$ vs. $g(x^{adv})$). Our approach differs from these existing approaches by leveraging the nonuniform changes among different DNN layer activations. Instead of analyzing the input $x^{adv}$ or the output $g(x^{adv})$, the proposed defense method analyzes the intermediate steps between $x^{adv}$ and $g(x^{adv})$.

To develop a universal detector that works with any DNN and against any attack, we start with the following generic observation. Although there are numerous attacks with different approaches to generate adversarial examples, all attacks essentially aim to change the model's prediction by maximizing the loss $\mathcal{L}$ (e.g., cross-entropy loss) between prediction $g(x^{adv})$ and the one-hot encoded ground truth $y$ while limiting the perturbation Fang et al. (2024):

$$\max_{x^{adv}} \mathcal{L}(g(x^{adv}), y) \;\; \text{s.t.} \;\; \|x^{adv} - x\|_{\infty} \leq \epsilon. \tag{1}$$

Considering this common aim of attack methods, "start with a small perturbation and end up with a big one", and the sequential nature of DNNs, we hypothesize that the impact of adversarial examples on the final layer

is higher than the initial layer. Let us first define a generic DNN $g(\cdot)$ consisting of $n$ layers $a = \{a_1, a_2, ..., a_n\}$. In DNNs, including CNNs, transformers, etc., layers incrementally process the information from the previous layers to compute their respective outputs to the next layer. For example, for a model $g$ where each layer is connected to the previous one, the final output of the model can be formulated as follows:

$$g(x) = a_n(a_{n-1}(\ldots a_1(x))). \tag{2}$$

For simplicity, we will denote a layer's output vector with $a_i(x)$. Note that, the output of the last layer $a_n(x) = g(x)$ is typically the class probability vector in classification tasks, and $a_{n-1}(x)$ is referred as the feature vector of the model.

Our Conjecture and Remark below to build a universal and efficient detector are based on the assumption that, for an adversarial sample $x^{adv}$, the first layer output $a_1(x^{adv})$ remains close to the clean version $a_1(x)$ since the change in the input is required to be unnoticeable by design, i.e., $\|x^{adv} - x\|_\infty \leq \epsilon$.

***Conjecture:*** Assume a loss function $\mathcal{L}(g(x), y)$ that is monotonic with $\|g(x) - y\|_\infty$. Assuming a highly accurate target model $g$ that is trained by optimizing the weights $w$ to effectively minimize the loss $\mathcal{L}(g_w(x), y)$, i.e.,

$$g(x) = g_{w^*}(x), \text{ where } w^* = \arg\min_w \mathcal{L}(g_w(x), y), \tag{3}$$

we can rewrite Equation 1 as

$$\max_{x^{adv}} \mathcal{L}(g(x^{adv}), g(x)) \text{ s.t. } \|x^{adv} - x\|_\infty \leq \epsilon. \tag{4}$$

Note that $a_n(x) = g(x)$, $a_n(x^{adv}) = g(x^{adv})$, and $\mathcal{L}(g(x^{adv}), g(x))$ is monotonic with $\|g(x^{adv}) - g(x)\|_\infty$. Hence, $\|a_n(x^{adv}) - a_n(x)\|_\infty$ is maximized while limiting $\|x^{adv} - x\|_\infty$ by a small number.

*Since the first layer is right next to the input, for which the perturbation is minimized, and far away from the final layer, for which the impact is maximized, we conjecture that, for $n > 1$,*

$$d_1 = \|a_1(x^{adv}) - a_1(x)\|_\infty < d_n = \|a_n(x^{adv}) - a_n(x)\|_\infty. \tag{5}$$

According to this conjecture, the impact of an adversarial sample is higher on the final layer output than the first layer output. We next present how to utilize this for detecting adversarial samples.

***Remark:*** Consider a function $f$ which maps $a_1(x)$ to $a_n(x)$ and is stable in the sense that $\|f(a_1(x)) - f(a_1(x+\epsilon))\|_\infty \leq \delta$ for small $\epsilon$ and $\delta$. Since $\|x^{adv} - x\|_\infty \leq \epsilon$ from Equation 1, assuming $a_1(x^{adv})$ stays close to $a_1(x)$, $f(a_1(x^{adv}))$ is close to $f(a_1(x))$ due to the stability of $f$. From Equation 5, $a_n(x^{adv})$ is far away from $a_n(x)$ compared to the distance between $a_1(x)$ and $a_1(x^{adv})$. Thus, the estimation error for adversarial samples is larger than the error for clean samples:

$$e_a = \|f(a_1(x^{adv})) - a_n(x^{adv})\|_\infty > e_c = \|f(a_1(x)) - a_n(x)\|_\infty. \tag{6}$$

### 3.1 Layer Regression

While Equation 5 provides the theoretical motivation, the result in Equation 6 offers a mechanism to detect adversarial samples if we can train a suitable function $f$ (Figure 1). We make four approximations to obtain a practical algorithm based on Equation 6. First, for computational efficiency and real-time detection even in resource-constrained systems, we propose to use a multi-layer perceptron (MLP) to approximate $f$. Second, since $a_n(x)$ denotes the predicted class probabilities, we choose the feature vector $a_{n-1}$, which takes unconstrained real values, as the target to train MLP as a regression model. Third, we approximate non-differentiable $\|\cdot\|_\infty$ with $\|\cdot\|_2$ to train the MLP using the differentiable mean squared error (MSE) loss. To empirically validate of Equation 5 under these three approximations, we use the ImageNet validation dataset, ResNet-50 He et al. (2015) as target model, and PGD attack Madry et al. (2017) to compute the mean and standard deviation of normalized change in layer 1, $\tilde{d}_1 = \frac{\|a_1(x^{adv}) - a_1(x)\|_2}{\|a_1(x^{adv})\|_2 + \|a_1(x)\|_2}$, and layer $n-1$, $\tilde{d}_{n-1} = \frac{\|a_{n-1}(x^{adv}) - a_{n-1}(x)\|_2}{\|a_{n-1}(x^{adv})\|_2 + \|a_{n-1}(x)\|_2}$.

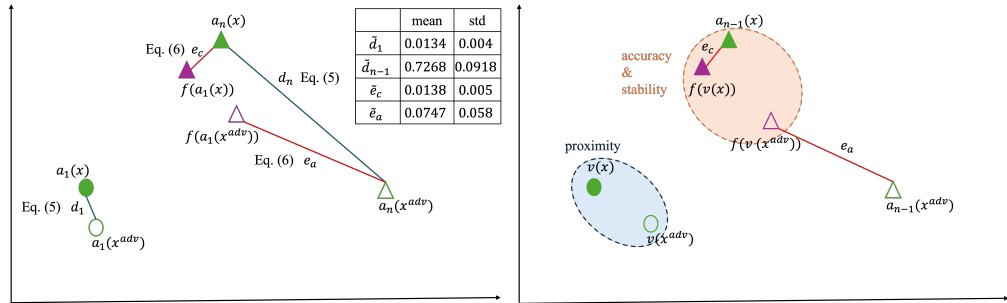

Figure 1: (Left) Equation 5 conjectures that the impact of adversarial samples is higher on the final layer than the first layer. Equation 6 uses this to show that the error of a stable estimator is higher for adversarial samples compared to clean samples. (Right) The performance of approximations in proposed detector to Equation 6 depend on two conflicting objectives: proximity of input vectors for clean and adversarial samples, and training an accurate and stable estimator. (Table) Empirical confirmation of the agreement between the proposed detector with approximations and the theoretical arguments.

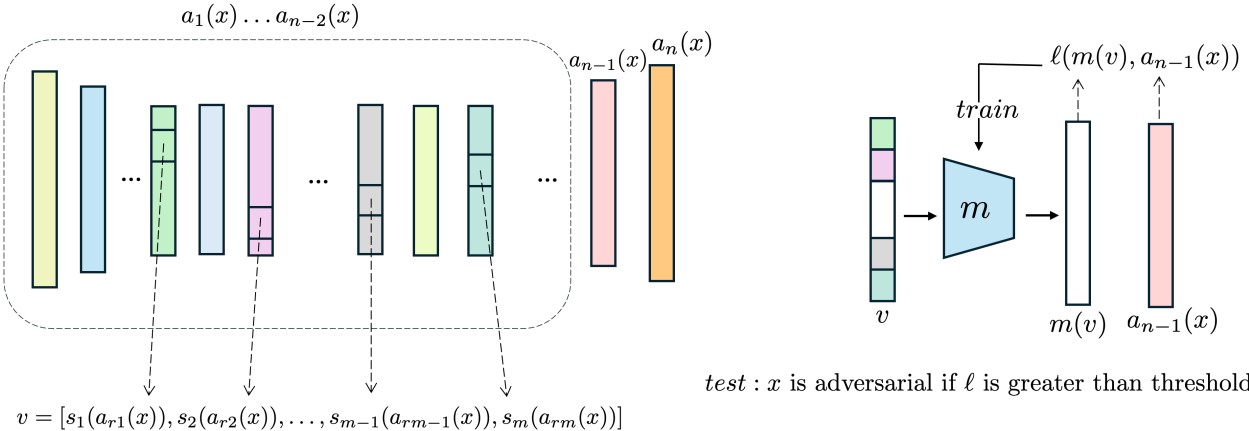

Figure 2: (Left) Layer selection and slicing operations to form the input vector. (Right) Training and testing procedures of the proposed detector.

In deep neural networks with $n \gg 1$, training a suitable $f$ to estimate the feature vector $a_{n-1}$ using the first layer output $a_1$ as the input is a challenging task due to the highly nonlinear mapping in $n-2$ layers. To develop a lightweight detector via MLP, as the fourth approximation, we propose selecting a mixture of early layer outputs as the input to the regression model instead of using only the first layer. Using a mixture of 5th, 8th, and 13th convolutional layers in ResNet-50 as the input, we empirically check Equation 6 under the same setting used for $\tilde{d}_1$ and $\tilde{d}_{n-1}$ by computing the mean and standard deviation of MSE $\tilde{e}_c = \|f(a_1(x)) - a_{n-1}(x)\|_2$ for clean images and $\tilde{e}_a = \|f(a_1(x^{adv})) - a_{n-1}(x^{adv})\|_2$ for adversarial images, where an MLP with two hidden layers is used for $f$. Results shown in Figure 1 corroborate Equation 5 and Equation 6 under the approximations. Input selection for MLP is further discussed in this section and in Supplementary D.

Utilizing the four approximations to Equation 6 discussed above, we propose a universal and lightweight detection algorithm, as shown in Figure 2, with the following steps: (i) select a subset of the first $n-2$ layer vectors and generate a new vector $v$ from the selected subset, (ii) feed $v$ into a regression model $m$ to predict the feature vector $a_{n-1}(x)$, (iii) train $m$ by minimizing the mean squared error (MSE) loss $\ell(m(v), a_{n-1}(x))$ in the clean training set devoid of adversarial samples, (iv) use $\ell(m(v), a_{n-1}(x))$ as the detection score. $m$ is expected to produce low scores for clean inputs and high for adversarial inputs. A pseudo-code of LR is given in Supplementary G.

The formation of vector $v$ can be formed in various ways, such as using only the $i$th layer vector $v = a_i(x)$ or mixture of several layer vectors. To enable larger estimation error $e_a$ for adversarial samples than estimation error $e_c$ for clean samples, the choice for $v$ should strike a balance between two competing goals, as illustrated in Figure 1: proximity of clean $v(x)$ and adversarial $v(x^{adv})$, and accuracy and stability of estimation function $f$. While training an accurate and stable regression model is more feasible when $v$ is selected from the layers closer to the target layer $n-1$, e.g., $v = a_{n-2}(x)$, such a detector might be less sensitive to adversarial samples since both $a_{n-2}(x)$ and $a_{n-1}(x)$ are expected to be impacted significantly by the attack, i.e., $a_{n-2}(x)$ and $a_{n-2}(x^{adv})$ will not be proximal. On the other hand, while selecting $v = a_1(x)$ ensures a reasonably small perturbation in $v$, it also makes obtaining an accurate and stable estimator more challenging. As a result, we propose to select a subset of layer vectors

$$a_r = \{a_{r1}(x), a_{r2}(x), ..., a_{rm}(x)\} \tag{7}$$

where $a_r \in a$ and $m < n$ is the number of selected layers. From the selected layer vectors, we aim to generate a new vector $v$. However, since the layer vectors are often large due to the operations like convolutions or attentions, to get a specific portion of the selected layer vectors, we define a unique slicing function $s = \{s_1, s_2, ..., s_m\}$ for each layer vector in $a_r$. Then, each slicing function is applied to the corresponding layer vector in $a_r$ to get the sliced vectors

$$s_r = \{s_1(a_{r1}(x)), s_2(a_{r2}(x)), ..., s_m(a_{rm}(x))\}. \tag{8}$$

Finally, the vector $v$ is generated by concatenating the vectors in $s_r$:

$$v = [s_1(a_{r1}(x)), s_2(a_{r2}(x)), \ldots, s_m(a_{rm}(x))]. \tag{9}$$

The order of $s_i(a_{ri}(x))$ in $v$ can be randomized in inference to counteract adaptive attacks, as explained in Section 4.4. The proposed layer selection and slicing process is summarized in Figure 2. During the training, only the clean input samples are used. After the training, the loss is expected to be low for clean inputs and high for adversarial inputs. The implementation of Layer Regression is publicly available at https://github.com/furkanmumcu/Layer-Regression.

## 4 Experiments

**Datasets and Evaluation:** For the evaluation on ImageNet, 10,000 images from the validation set were chosen. For CIFAR-100, the test set, which also contains 10,000 images, was used. The commonly used area under the receiver operating characteristic (AUROC) curve is used to evaluate the attack detection performance of defense methods. Similarly to existing work (Zhang et al., 2023), the evaluation is conducted on adversarial and clean sets. The clean set comprises all images that are correctly classified by the target models. An adversarial set is formed for each attack-target model combination by gathering the attack's adversarial images misclassified by target model. More details on the evaluation methods and datasets are given in Supplementary F.

**Baselines:** We benchmark our method against a diverse range of baseline approaches, including six popular computationally efficient methods: JPEG compression (JPEG) (Das et al., 2018), Randomization (Random)(Xie et al., 2017), Deflection (Deflect) (Prakash et al., 2018), Feature Squeezing (FS) Xu (2017a), Wavelet denoising (Denoise) and super resolution (WDSR) (Mustafa et al., 2019), and two most recent state-of-the-art detection methods: VLAD (Mumcu & Yilmaz, 2024c) and EPS-AD (Zhang et al., 2023). Since FS, VLAD and EPS-AD were proposed as detectors in their respective papers, we use their official implementations. The remaining methods were proposed to increase robustness by altering the inputs to remove perturbations. Therefore, we derive a detection method from them by comparing the predictions before and after the methods are applied. The case the predictions match indicates no attack and a mismatch indicates an attack.

**Target Models**: The experiments are conducted using the validation set of ImageNet dataset (Russakovsky et al., 2015) and the CIFAR-100 dataset (Krizhevsky et al., 2009). To represent the different architectures we used three CNN-based image classification models, VGG19 Simonyan & Zisserman (2014), ResNet50

He et al. (2015), InceptionV3 Szegedy et al. (2015); and three transformer-based models, ViT Dosovitskiy (2020), DeiT Touvron et al. (2021), LeViT Graham et al. (2021).

**Attack Methods:** In this section, we use the untargeted $l_\infty$ attack setting, however we also report our detector's performance under targeted and $l_2$ attack settings in Supplementary B. We consider two different threat models in the experiments. The first one is the white-box static attack setting in which the attacker has complete knowledge of the classifier but not the detector. Strong attacks from the literature, namely the BIM (Kurakin et al., 2018), PGD (Madry et al., 2017), PIF (Gao et al., 2020), APGD (Croce & Hein, 2020b), ANDA (Fang et al., 2024), VMI and VNI (Wang & He, 2021) attacks are used as white-box attacks. We also test against an ensemble white-box attack, called AutoAttack (AA) (Croce & Hein, 2020a).

The second threat model is the adaptive attack setting in which the attacker also has the full knowledge of defense mechanism. We analyze the performance of our detector against a specially crafted adaptive attack Yang et al. (2022) which is trained to simultaneously deceive the target model and bypass our detector in Section 4.4.

**LR training:** For each model, an MLP with 2 hidden layers is trained as the LR detector to demonstrate that a highly effective detector can be built using a very lightweight neural network with minimal computational overhead. After experimenting with different layer selection and slicing strategies, we found that selecting only from early or final layers reduce the performance, but randomly selecting three layers and slicing the middle 60% portion of each selected layer yields good results. The selection procedure of layers $a_r$ and slicing functions $s_r$, and training parameters are detailed in Supplementary F.

## 4.1 Detecting White-Box Static Attacks

We conducted extensive comparisons with the existing computationally-efficient defenses. In Table 1 we report the AUROC scores for JPEG, Random, Deflect, Denoise, WDSR, FS and our method against BIM, PGD, PIF, APGD, ANDA, VMI and VNI attacks targeting 7 models, namely VGG19, ResNet50, InceptionV3, ViT, DeiT, LeViT, on two datasets, ImageNet and CIFAR-100. In every experimental setting, with different attacks, target models, and datasets, our proposed method outperforms these defense methods by a wide margin. Compared to LR's average AUROC score of 0.98, the best performance among the existing efficient methods remains at 0.64. More importantly, our method is robust across varying target models and attack types, as indicated by its small standard deviation. While the other defense methods are effective against certain target-attack combinations, they fail to generalize this to a wide range of settings. For instance, on ImageNet, JPEG, Random, and FS perform better with transformer models, however they rarely exceed the random guess performance with the CNN models.

Due to the high computational requirements of VLAD and EPS-AD, we benchmark these methods in a smaller setting, where 1,000 images and 3 target models are used with the same 7 attacks and the results are given in Table 2. In all of the test cases, VLAD has an average AUROC score of 0.88. It performs worst against the attacks targeted ResNet50 where its performance varies between 0.82 and 0.77. On the other hand, similarly to the results demonstrated in Table 1, LR proves its robustness to different attack and target combinations with the average AUROC score of 0.99. EPS-AD has a similar performance to our method where it successfully detects attacks with an average AUROC score of 0.99. However, compared to LR, both VLAD and EPS-AD have considerably large computational cost, which limits their real-world usage. In the next section, we discuss the computational efficiency and real-world applicability of all defense methods considered in this section.

## 4.2 Computational Efficiency

Real-time attack detection is a crucial aspect of many real-world systems. A detection mechanism must operate consistently alongside the DNN model to ensure timely identification of adversarial examples. Consequently, an ideal detector should be computationally efficient. In this section, we compare the computational costs of the defense methods evaluated in our experiments. For each defense method, we processed 1,000 samples from ImageNet and calculated the average processing time per sample. Figure 3 presents the processing time per sample (PTS) in seconds vs. AUROC for each defense method. Our method is

| | | ImageNet | | | | | | | CIFAR-100 | | | | | | |
|---|---|---|---|---|---|---|---|---|---|---|---|---|---|---|---|
| | | JPEG | Random | Deflect | Denoise | WDSR | FS | LR (Ours) | JPEG | Random | Deflect | Denoise | WDSR | FS | LR (Ours) |
| VGG19 | BIM | 0.42 | 0.43 | 0.48 | 0.45 | 0.39 | 0.13 | **0.99** | 0.47 | 0.31 | 0.49 | 0.48 | 0.41 | 0.01 | **0.99** |
| | PGD | 0.50 | 0.46 | 0.48 | 0.45 | 0.46 | 0.22 | **0.99** | 0.48 | 0.31 | 0.49 | 0.48 | 0.44 | 0.02 | **0.99** |
| | PIF | 0.33 | 0.44 | 0.48 | 0.45 | 0.31 | 0.08 | **0.99** | 0.47 | 0.31 | 0.49 | 0.48 | 0.41 | 0.01 | **0.99** |
| | APGD | 0.53 | 0.46 | 0.48 | 0.45 | 0.49 | 0.23 | **0.99** | 0.48 | 0.32 | 0.49 | 0.48 | 0.42 | 0.02 | **0.99** |
| | ANDA | 0.39 | 0.47 | 0.49 | 0.47 | 0.38 | 0.26 | **0.95** | 0.48 | 0.36 | 0.49 | 0.49 | 0.43 | 0.11 | **0.99** |
| | VMI | 0.36 | 0.42 | 0.48 | 0.45 | 0.34 | 0.11 | **0.99** | 0.47 | 0.32 | 0.49 | 0.48 | 0.41 | 0.01 | **0.99** |
| | VNI | 0.41 | 0.45 | 0.48 | 0.46 | 0.38 | 0.19 | **0.99** | 0.47 | 0.32 | 0.49 | 0.48 | 0.42 | 0.02 | **0.99** |
| ResNet50 | BIM | 0.67 | 0.63 | 0.49 | 0.48 | 0.55 | 0.29 | **0.99** | 0.82 | 0.69 | 0.46 | 0.62 | 0.73 | 0.39 | **0.98** |
| | PGD | 0.77 | 0.75 | 0.49 | 0.49 | 0.68 | 0.53 | **0.98** | 0.87 | 0.71 | 0.45 | 0.71 | 0.82 | 0.40 | **0.99** |
| | PIF | 0.51 | 0.64 | 0.49 | 0.48 | 0.47 | 0.29 | **0.96** | 0.73 | 0.72 | 0.45 | 0.48 | 0.61 | 0.63 | **0.97** |
| | APGD | 0.75 | 0.70 | 0.49 | 0.49 | 0.64 | 0.47 | **0.97** | 0.84 | 0.65 | 0.46 | 0.62 | 0.78 | 0.44 | **0.96** |
| | ANDA | 0.45 | 0.48 | 0.49 | 0.48 | 0.44 | 0.14 | **0.96** | 0.56 | 0.53 | 0.45 | 0.48 | 0.45 | 0.07 | **0.99** |
| | VMI | 0.49 | 0.52 | 0.49 | 0.48 | 0.45 | 0.14 | **0.99** | 0.61 | 0.58 | 0.45 | 0.50 | 0.50 | 0.23 | **0.99** |
| | VNI | 0.53 | 0.54 | 0.49 | 0.49 | 0.47 | 0.21 | **0.97** | 0.62 | 0.55 | 0.45 | 0.50 | 0.51 | 0.28 | **0.97** |
| InceptionV3 | BIM | 0.55 | 0.53 | 0.50 | 0.49 | 0.77 | 0.24 | **0.98** | 0.88 | 0.80 | 0.43 | 0.74 | 0.81 | 0.36 | **1.00** |
| | PGD | 0.62 | 0.58 | 0.50 | 0.49 | 0.80 | 0.36 | **0.97** | 0.86 | 0.66 | 0.43 | 0.76 | 0.81 | 0.32 | **0.99** |
| | PIF | 0.48 | 0.52 | 0.50 | 0.49 | 0.67 | 0.11 | **0.99** | 0.82 | 0.89 | 0.44 | 0.57 | 0.72 | 0.34 | **1.00** |
| | APGD | 0.60 | 0.56 | 0.50 | 0.50 | 0.79 | 0.36 | **0.96** | 0.88 | 0.82 | 0.43 | 0.77 | 0.81 | 0.38 | **0.99** |
| | ANDA | 0.49 | 0.50 | 0.50 | 0.49 | 0.52 | 0.23 | **0.92** | 0.71 | 0.57 | 0.43 | 0.50 | 0.59 | 0.29 | **0.99** |
| | VMI | 0.48 | 0.50 | 0.50 | 0.49 | 0.67 | 0.12 | **0.98** | 0.85 | 0.78 | 0.42 | 0.50 | 0.80 | 0.35 | **1.00** |
| | VNI | 0.52 | 0.53 | 0.50 | 0.50 | 0.70 | 0.25 | **0.96** | 0.86 | 0.79 | 0.43 | 0.60 | 0.81 | 0.31 | **1.00** |
| ViT | BIM | 0.87 | 0.90 | 0.53 | 0.64 | 0.82 | 0.93 | **0.99** | 0.50 | 0.66 | 0.50 | 0.50 | 0.49 | 0.29 | **0.98** |
| | PGD | 0.85 | 0.87 | 0.53 | 0.60 | 0.79 | 0.91 | **0.99** | 0.49 | 0.67 | 0.50 | 0.50 | 0.48 | 0.27 | **0.97** |
| | PIF | 0.79 | 0.82 | 0.52 | 0.51 | 0.67 | 0.87 | **0.99** | 0.49 | 0.53 | 0.50 | 0.49 | 0.48 | 0.15 | **0.98** |
| | APGD | 0.86 | 0.90 | 0.53 | 0.63 | 0.80 | 0.92 | **0.99** | 0.50 | 0.71 | 0.50 | 0.50 | 0.50 | 0.32 | **0.95** |
| | ANDA | 0.72 | 0.68 | 0.52 | 0.56 | 0.64 | 0.82 | **0.97** | 0.49 | 0.52 | 0.50 | 0.50 | 0.48 | 0.26 | **0.85** |
| | VMI | 0.77 | 0.81 | 0.51 | 0.57 | 0.70 | 0.86 | **0.99** | 0.49 | 0.58 | 0.50 | 0.49 | 0.48 | 0.15 | **0.99** |
| | VNI | 0.78 | 0.83 | 0.52 | 0.61 | 0.73 | 0.91 | **0.99** | 0.50 | 0.60 | 0.50 | 0.50 | 0.49 | 0.27 | **0.95** |
| DeiT | BIM | 0.86 | 0.89 | 0.52 | 0.58 | 0.78 | 0.88 | **0.99** | 0.50 | 0.64 | 0.50 | 0.49 | 0.48 | 0.36 | **0.92** |
| | PGD | 0.86 | 0.88 | 0.53 | 0.58 | 0.80 | 0.90 | **0.99** | 0.50 | 0.66 | 0.50 | 0.49 | 0.48 | 0.34 | **0.87** |
| | PIF | 0.77 | 0.84 | 0.51 | 0.50 | 0.71 | 0.62 | **0.99** | 0.48 | 0.52 | 0.50 | 0.49 | 0.47 | 0.16 | **0.97** |
| | APGD | 0.85 | 0.90 | 0.52 | 0.57 | 0.78 | 0.88 | **0.99** | 0.50 | 0.65 | 0.50 | 0.50 | 0.48 | 0.36 | **0.91** |
| | ANDA | 0.77 | 0.76 | 0.53 | 0.58 | 0.70 | 0.75 | **0.99** | 0.49 | 0.51 | 0.50 | 0.49 | 0.48 | 0.26 | **0.91** |
| | VMI | 0.79 | 0.84 | 0.51 | 0.53 | 0.69 | 0.80 | **0.99** | 0.49 | 0.56 | 0.50 | 0.49 | 0.48 | 0.30 | **0.96** |
| | VNI | 0.80 | 0.85 | 0.52 | 0.55 | 0.72 | 0.85 | **0.99** | 0.49 | 0.57 | 0.50 | 0.50 | 0.48 | 0.28 | **0.94** |
| LeViT | BIM | 0.68 | 0.73 | 0.50 | 0.50 | 0.62 | 0.64 | **0.99** | 0.93 | 0.83 | 0.68 | 0.88 | 0.86 | 0.98 | **0.99** |
| | PGD | 0.69 | 0.74 | 0.50 | 0.49 | 0.62 | 0.66 | **0.99** | 0.93 | 0.84 | 0.57 | 0.84 | 0.85 | 0.86 | **0.93** |
| | PIF | 0.54 | 0.65 | 0.50 | 0.49 | 0.49 | 0.51 | **0.99** | 0.86 | 0.82 | 0.50 | 0.53 | 0.80 | 0.62 | **0.93** |
| | APGD | 0.75 | 0.80 | 0.50 | 0.51 | 0.69 | 0.76 | **0.98** | 0.93 | 0.83 | 0.68 | 0.88 | 0.86 | 0.98 | **0.99** |
| | ANDA | 0.49 | 0.51 | 0.50 | 0.49 | 0.49 | 0.43 | **0.94** | 0.84 | 0.71 | 0.52 | 0.57 | 0.81 | 0.65 | **0.93** |
| | VMI | 0.53 | 0.59 | 0.50 | 0.49 | 0.48 | 0.39 | **0.99** | 0.93 | 0.83 | 0.66 | 0.66 | 0.86 | 0.98 | **0.99** |
| | VNI | 0.60 | 0.66 | 0.50 | 0.51 | 0.54 | 0.59 | **0.99** | 0.92 | 0.83 | 0.67 | 0.71 | 0.86 | 0.99 | **0.99** |
| Average | | 0.63 | 0.66 | 0.50 | 0.51 | 0.61 | 0.50 | **0.99** | 0.65 | 0.62 | 0.50 | 0.56 | 0.60 | 0.35 | **0.97** |

Table 1: Comparison between 7 computationally efficient defenses in terms of detection AUROC against 7 attack methods targeting 6 models using ImageNet and CIFAR-100 datasets.

| | | BIM | PGD | PIF | APGD | ANDA | VMI | VNI |
|---|---|---|---|---|---|---|---|---|
| VGG19 | VLAD | 0.94 | 0.94 | 0.96 | 0.96 | 0.93 | 0.93 | 0.95 |
| | EPS-AD | **0.99** | 0.97 | 0.97 | **0.99** | 0.98 | **0.99** | **0.99** |
| | LR (Ours) | **0.99** | **0.99** | **0.99** | **0.99** | **0.99** | **0.99** | **0.99** |
| ResNet | VLAD | 0.81 | 0.81 | 0.81 | 0.82 | 0.77 | 0.80 | 0.80 |
| | EPS-AD | **0.99** | 0.98 | **0.98** | **0.99** | **0.99** | **0.99** | **0.99** |
| | LR (Ours) | **0.99** | **0.99** | **0.98** | 0.97 | 0.96 | **0.99** | 0.97 |
| ViT | VLAD | 0.94 | 0.92 | 0.89 | 0.93 | 0.85 | 0.93 | 0.91 |
| | EPS-AD | **0.99** | 0.97 | 0.98 | **0.99** | **0.99** | **0.99** | **0.99** |
| | LR (Ours) | **0.99** | **0.99** | **0.99** | **0.99** | **0.99** | **0.99** | **0.99** |

Table 2: Comparison between state-of-the-art defense methods in terms of detection AUROC on ImageNet.

the fastest, with a processing time of just 0.0004 seconds. In contrast, VLAD, WSDR, and EPS-AD are the slowest, with PTS values of 0.1431, 0.2611, and 2.9998 seconds, respectively. Our method demonstrates exceptional detection performance and is also the fastest. Notably, our method runs $1.3 \times 10^6$ times faster than EPS-AD, which achieves similar detection performance to ours. The processing times are measured using a desktop computer with an NVIDIA 4090 GPU, AMD Ryzen 9 7950X CPU, and 64 GB RAM.

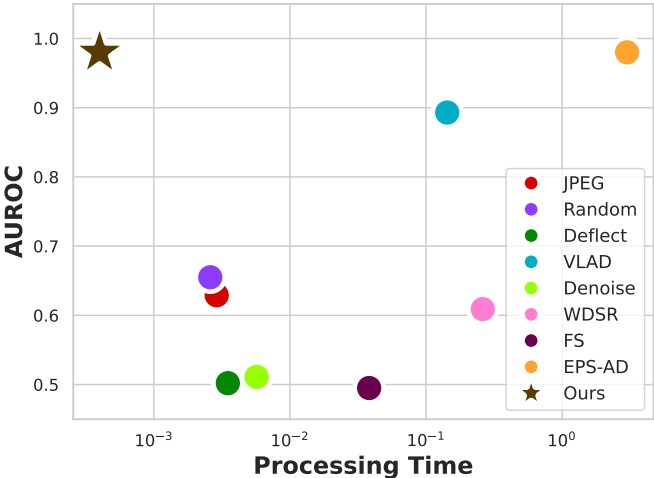

Figure 3: The proposed detector is both universal (high AUROC across all scenarios) and lightweight (fastest among all existing defenses).

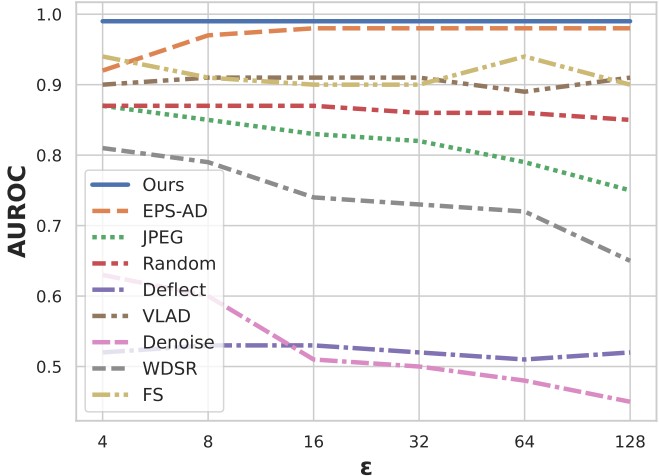

Figure 4: Detection performance vs attack strength.

### 4.3 Performance under Varying Attack Strength

Adversarial attacks often use a parameter $\epsilon$ to adjust the amount of perturbation on adversarial examples. For the experiments in Section 4, we used the default attack settings that are proposed by the authors. In this study, we investigate the performance of defense methods against PGD attack with different $\epsilon$ values. In the original implementation of PGD, attack strength parameter $\epsilon$ is set to 8. In addition to the original $\epsilon$, we also generated attacks by setting $\epsilon$ to 4, 16, 32, 64 and 128. Figure 4 plots the performances of defense methods for ViT-PGD model-attack pair. Our method performs the same for every $\epsilon$ value, since LR depends on the perturbation's effects on DNN layers, a significant performance drop is not expected due to the changes in attack strength. EPS-AD performs lower on weaker attacks, especially when $\epsilon$ is set to 4, their performance drops considerably. In remaining defense methods, while VLAD and Deflect experience only small changes under stronger attacks, whereas the others are greatly affected by the $\epsilon$ value.

| Model | VGG19 | ResNet | IncV3 | ViT | DeiT | LeViT |
|-------|-------|--------|-------|-----|------|-------|
| AUROC | 0.99 | 0.97 | 0.96 | 0.99 | 0.99 | 0.98 |

Table 3: LR's performance against AutoAttack.

| $\epsilon$ | 4 | 8 | 16 | 32 | 64 | 128 |
|-----------|-----|-----|-----|-----|-----|-----|
| AUROC | 0.81 | 0.78 | 0.79 | 0.80 | 0.81 | 0.81 |

Table 4: LR's performance under varying strengts of PGD-based adaptive attack where $\lambda$ is set to 1.

### 4.4 Performance against Adaptive Attacks

AutoAttack is an ensemble-based method combining multiple attacks. In recent works, it is used as an adaptive atttack (Zhang et al., 2023) to evaluate detectors. In Table 3, we provide LR's performance against AutoAttack targetting 6 models. Similarly to the results in Section 4.1, LR performs with an average of 0.98% AUROC score.

Since AutoAttack does not use any knowledge of the detector, we design a PGD-based targeted adaptive attack, following a similar approach to (Yang et al., 2022), with the objective function

$$\max L_{Classifier} - \lambda \cdot L_{LR}, \tag{10}$$

where $L_{Classifier}$ and $L_{LR}$ denote the loss functions of the target model and LR detector, respectively. The sign of LR's loss function is (-) since LR identifies an input as clean when the loss is low. Note that, since this threat model has the complete knowledge of the detector, it is extremely challenging and destructive for the detector. To cope with this challenge, we propose randomly concatenating the layer vectors described in Equation equation 9 at inference time.

In Table 4, with ViT as the target model and the experimental settings proposed in (Yang et al., 2022) where number of iterations are set as 200 and $\lambda$ is set as 1, we report LR's performance under adaptive PGD attack with varying attack budgets $\epsilon \in [4, 8, 16, 32, 64, 128]$. LR achieves an average AUROC score of 80%, even under this challenging adaptive threat model

## 5 Applicability in Other Domains

LR is a universal detector applicable in various domains where DNNs are used. Here, we demonstrate its performance in two additional domains, video action recognition and speech recognition. In Supplementary A and I, we provide more information about the experimental details used in this section. We also present additional results on another application, traffic sign recognition, in Supplementary H.

### 5.1 LR for Video Action Recognition

In this section, we implement LR against video action recognition attacks and compare its performance to existing defenses designed for action recognition models, namely Advit (Xiao et al., 2019), Shuffle (Hwang et al., 2023), and VLAD (Mumcu & Yilmaz, 2024c). In the experiments, PGD-v attack (Mumcu & Yilmaz, 2024c) and Flick attack (Pony et al., 2021) are used to target MVIT (Fan et al., 2021) and X3D (Feichtenhofer, 2020). The experimental settings in Mumcu & Yilmaz (2024c) on Kinetics-400 (Kay et al., 2017) dataset are followed. The results in Table 5 show that LR outperforms the other defenses with an average AUROC of 0.93%, followed by VLAD which achieves 0.91%.

### 5.2 LR for Speech Recognition

To demonstrate LR's wide applicability beyond computer vision, we also evaluate its performance against a speech recognition attack. As target model, we use Wav2vec (Schneider et al., 2019) model, which is trained on LibriSpeech (Panayotov et al., 2015) dataset. Model is attacked with FGSM (Goodfellow et al., 2014). LR achieves an average AUROC of 0.99. Figure 5 illustrates some attacked and clean samples from LibriSpeech,

|  |  | Advit | Shuffle | VLAD | LR (ours) |
|---|---|---|---|---|---|
| MVIT | PGD-v | 0.93 | 0.98 | 0.93 | **0.99** |
|  | Flick | 0.34 | 0.65 | 0.87 | **0.89** |
| X3D | PGD-v | 0.92 | 0.76 | **0.97** | 0.95 |
|  | Flick | 0.54 | 0.59 | 0.90 | **0.92** |
|  | Average | 0.68 | 0.74 | 0.91 | **0.93** |

Table 5: AUROC scores against attack methods targeting video action recognition models.

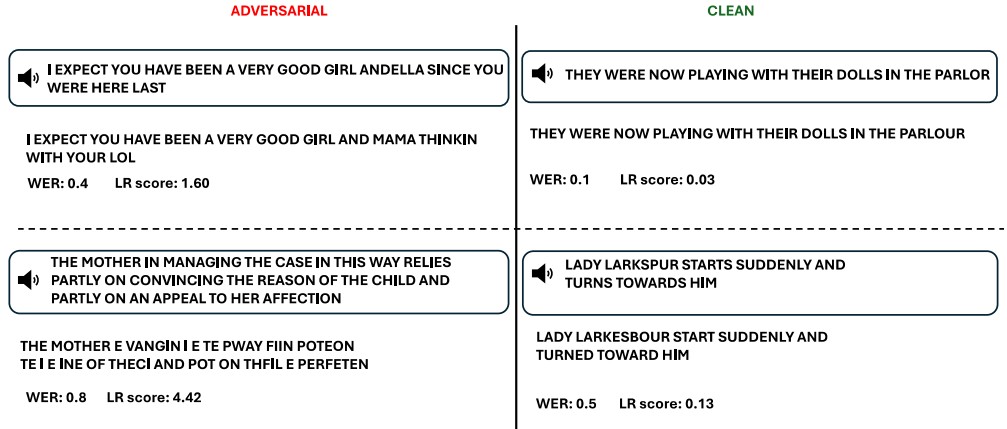

Figure 5: Clean and adversarial samples for speech recognition with ground truth in box, recognized text, word error rate (WER), and MSE of proposed LR.

along with the MSE values of LR. As shown in the figure, LR can even detect stealthy attacks which cause minimal increase in word error rate (WER) while distorting the recognized speech. Remarkably, while the WER of the first adversarial example is lower than that of the second clean example (0.4 vs. 0.5), the LR loss for this stealthy adversarial sample is more than ten folds greater than that of the second clean example (1.6 vs 0.13).

# 6  Conclusion

Although there are effective defense methods for specific model-attack combinations, their success does not generalize to all popular models and attacks. In this work, we filled this gap by introducing Layer Regression (LR), the first universal method for detecting adversarial examples. In addition to its universality, LR is much faster than the existing defense methods. By analyzing the common objectives of attacks and the sequential layer-based nature of DNNs, we showed that adversarial samples have greater impact on the final layer than on the initial layers. Motivated by this, LR trains a lightweight multi-layer perceptron (MLP) on clean samples to estimate the feature vector using a combination of outputs from early layers. The estimation error of LR for adversarial samples is typically much higher than the error for clean samples. Through extensive experiments, we showed that LR outperforms the existing defenses in image recognition by a wide margin in terms of speed and detection performance, and also delivers highly effective results in distinct domains, namely video action recognition and speech recognition. While inference time randomization of input order in LR is shown to be a promising technique for addressing adaptive attacks that are fully aware of LR, further research is needed to explore its full potential in this challenging scenario.

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

## A    More Related Work from Other Domains

Adversarial examples are studied on several other domains including video action recognition (Pony et al., 2021), traffic sign recognition (Hsiao et al., 2024), object detection (Zhao et al., 2019), video anomaly detection (Mumcu et al., 2022), and speech recognition (Żelasko et al., 2021). For instance, flickering attack (Flick) (Pony et al., 2021) tries to attack action recognition models by changing the RGB stream of the input videos. Hsiao et al. (2024) investigates the effects of natural light on traffic signs and use it to generate adversarial examples.

Advit (Xiao et al., 2019) is a detection method introduced for videos. It generates pseudo frames using optical flow and evaluates the consistency between the outputs for original inputs and pseudo frames to detect attacks. Another defense method, Shuffle (Hwang et al., 2023), tries to increase the robustness of action recognition models by randomly shuffling the input frames. (Mumcu & Yilmaz, 2024b) proposed to train a lightweight VLM and use it for adversarial traffic sign detection. In speech recognition, defenses like denoising or smoothing are studied against adversarial examples (Żelasko et al., 2021).

## B    LR's Performance Against $l_2$ and Targeted Attacks

|  | $PGD_{targeted}$ | $PGD_{l_2}$ | $APGD_{targeted}$ | $APGD_{l_2}$ |
|---|---|---|---|---|
| ResNet50 | 0.99 | 0.95 | 0.98 | 0.95 |
| ViT | 0.99 | 0.95 | 0.98 | 0.96 |

Table 6: LR's performance under $l_2$ and targeted attack settings.

We evaluate LR's performance under $l_2$ and targeted attack settings, specifically against $PGD_{targeted}$, $PGD_{l_2}$, $APGD_{targeted}$ and $APGD_{l_2}$ attacks using two different target models ResNet50 and ViT on ImageNet. The results are reported in Table 6. Similar to the white-box attack setting, LR performs an average AUROC score of 97% which further demonstrates the success of our detection method in different types of attacks.

## C    More Details on Experimental Settings

From the validation set of ImageNet (Russakovsky et al., 2015), we used 40,000 images to train the detectors and the remaining 10,000 for testing. In CIFAR-100 (Krizhevsky et al., 2009), there are 100 classes, 500 training images and 100 testing images per class, resulting in a total of 50,000 training and 10,000 test images. During the tests, only the correctly classified images by the target models were used. Total number for test images for each specific model and dataset is given in Table 7.

|  | VGG19 | ResNet50 | InceptionV3 | ViT | DeiT | LeViT |
|---|---|---|---|---|---|---|
| ImageNet | 4257 | 7701 | 7410 | 8457 | 7907 | 7639 |
| CIFAR-100 | 7045 | 8315 | 8089 | 8544 | 8667 | 8359 |

Table 7: Number of test images used for each model.

VLAD (Mumcu & Yilmaz, 2024c) was originally proposed for video recognition attacks, with the initial implementation designed for videos consisting of 30 frames. In the experiments conducted in Section 4, we used the official VLAD implementation. However, instead of averaging the scores across 30 frames, we applied it to a single image.

## D    Effects of Layer Vector Choice

For the experiments in Section 4, we used three layer vectors to form $v$, i.e., $m = 3$ in Eq. 9. Since the number of possible combinations for layer selection and slicing creates an intractably large search space, we limited our optimization to a representative subset of options. Table 8 summarizes the results of a preliminary

experiment on CIFAR-100, which is conducted with DeiT as the target model and PGD and PIF as the adversarial attacks. In this experiment, we considered 25 attention layers of the model and trained our detectors by forming $v$ in four different ways: (1) using only the first attention layer vector $a_1$, (2) using only the last attention layer vector $a_{25}$, (3) combining the vectors from fifth, sixth, and seventh attention vectors $[a_5, a_6, a_7]$, (4) and combining the vectors from eighth, thirteenth, and seventeenth attention vectors $[a_8, a_{13}, a_{17}]$. The best strategy turns out to combining vectors from early layers $[a_5, a_6, a_7]$ as it strikes a good balance in the trade-off between the two conflicting goals (Figure 1): proximity of $v(x)$ and $v(x^{adv})$, and accuracy and stability of estimator $f$. Additionally, having multiple layers and slicing functions can create randomness in each LR implementation, which makes developing an attack against our detection system harder. The layer selection and slicing strategies that gave the best results reported in Table 1 are listed in Supplementary F.

| Attack | $a_1$ | $[a_5, a_6, a_7]$ | $[a_8, a_{13}, a_{17}]$ | $a_{n-2}$ |
|---|---|---|---|---|
| PGD | 0.697 | 0.867 | 0.751 | 0.567 |
| PIF | 0.744 | 0.972 | 0.746 | 0.521 |

Table 8: Impact of layer selection on detection score.

## E    Layer Vector Selection Procedure

According to our preliminary experiments, using vectors exclusively from either early or final layers negatively impacts performance. Therefore, we propose selecting three layer vectors randomly between 1/5 and 4/5 of all layers. The proposed procedure for selecting layer vectors is detailed in Algorithm 1.

---
**Algorithm 1** Layer Vector Selection Procedure

---
1: **Input:** all layer vectors $v_n$,
2: **Output:** selected layer vectors $v_m$.
3: Get the layer size of all layer vectors $v_n$ :
4: $n \leftarrow len(v_n)$
5: Decide upper and lower bounds:
6: $a \leftarrow n/5$
7: $b \leftarrow 4 \cdot n/5$
8: Slice the vectors between the upper and lower bounds:
9: $v_m \leftarrow v_n[a : b]$
10: Randomly pick 3 vectors from $v_m$:
11: $v_m \leftarrow random.choices[v_m, 3]$
12: Return $v_m$

---

## F    Selected Layer Vectors and Slicing Functions

A specific subset of layer vectors $a_r$, as described in Equation equation 7, is chosen for each target model that is used during the experiments in Section 4. For the models, Pytorch Image Models (timm) (Wightman, 2019) is used.

We used the following layer subsets for the target models:

1. Resnet50: Layers with the name *conv2* were filtered, then among 15 *conv2* layers, 5th, 8th and 13th layers, for all ImageNet and CIFAR-100 tests

2. InceptionV3: Layers with the name *conv* were filtered, then among 94 *conv* layers, 15th, 25th and 35th layers, for all ImageNet and CIFAR-100 tests

3. VGG19: Layers with the name *features* were filtered, then among 37 *features* layers, 8th, 13th and 17th layers, for all ImageNet and CIFAR-100 tests

4. ViT: Layers with the name *attn.proj* were filtered, then among 23 *attn.proj* layers, 8th, 13th and 17th layers, for all ImageNet and CIFAR-100 tests

5. DeiT: Layers with the name *attn.proj* were filtered, then among 24 *attn.proj* layers, 8th, 13th and 17th layers for ImageNet tests, 5th, 6th, 7th layers for CIFAR-100 tests

6. LeViT: Layers with the name *attn.proj* were filtered, then among 12 *attn.proj* layers, 3th, 5th and 7th layers for ImageNet tests, 5th, 6th, 7th layers for CIFAR-100 tests

Before concatenating the $a_r$ vectors, a specific slicing function for each vector is applied as described in Equation equation 8. The specific slicing functions for each corresponding $a_r$ are detailed below:

1. Resnet50: $[: 5, : 28, : 28]$, $[: 50, : 7, : 7]$ and $[: 10, : 14, : 14]$.

2. InceptionV3: $[: 3, : 35, : 35]$, $[3 :, 35 :, 35]$ and $[: 3, : 17, : 17]$.

3. Vgg19: $[: 5, : 25, : 25]$, $[: 5, : 25, : 25]$ and $[: 5, : 25, : 25]$.

4. ViT: $[:, : 4 : 200]$, $[:, : 4 : 200]$ and $[:, : 4 :, 200]$.

5. DeiT: $[:, : 4, : 200]$, $[:, : 4, : 200]$ and $[:, : 4, : 200]$.

6. LeViT $[: 4, : 14 :, 14]$, $[: 14, : 7, : 7]$ and $[: 14, : 7, : 7]$.

The vector $v$ is generated by concatenating the sliced $a_r$ vectors as described in Equation equation 9. After acquiring $v$ for a model, an MLP with two hidden layers is trained to minimize the MSE loss between $v$ and the feature vector $a_{n-1}$. Adam optimizer with $3 \cdot 10^{-4}$ learning rate is used for the training.

## G Pseudo-Code for LR

In Section 3.1, we explain our detection algorithm in detail. Here, in Algorithm 2, we also provide a PyTorch Paszke et al. (2017) like pseudo code for LR.

---

**Algorithm 2** Layer Regression (LR)

---

1: **Input:** input $x$, selected subset of vector layers $a_r$, slicing functions $s$, DNN model $g$, feature vector $a_{n-1}$, LR detector $m$.
2: **Output:** loss $l$ if not training mode.
3: DNN $g$ takes the input $x$ as $g(x)$
4: Each layer vector $a_r$, process with corresponding slicing function in $s$
5: **for** $a_i$ in $a_r$ **do**
6: $\quad s_r \leftarrow s_i(a_i(x))$
7: **end for**
8: $v \leftarrow torch.cat(s_r)$
9: feed $v$ into a detection model $m$, such that: $m(v)$
10: calculate the MSE loss between $m(v)$ and $a_{n-1}$:
11: $l \leftarrow MSE(m(v), a_{n-1})$
12: **if** training **then**
13: $\quad l.backward()$
14: $\quad optimizer.step()$
15: **else**
16: $\quad$ Return $l$
17: **end if**

---

## H    Additional Study: LR in Traffic Sign Detection

|     | FGSM | PGD | Patch | Light | Average |
|-----|------|-----|-------|-------|---------|
| LR  | 0.97 | 0.99 | 0.95 | 0.94 | 0.96 |

Table 9: Detection AUROC of LR against four attacks targeting ResNet50 in the traffic sign recognition task.

As an additional experiment, we implement LR against attacks that target traffic sign recognition. In this study, ResNet50 is used as target model and attacked with FGSM(Goodfellow et al., 2014), PGD(Madry et al., 2017), Light(Hsiao et al., 2024) and Patch (Ye et al., 2021) attacks. In Table 9, we show that LR achieves an average AUC score of 96%, which further proves the applicability and success of our detection method across different domains.

## I    Details of Ablation Study & Other Domain Experiments

For Section 5.1, the experimental settings detailed in Mumcu & Yilmaz (2024c) is followed: "A subset of Kinetics-400 (Kay et al., 2017) is randomly selected for each target model from the videos that are correctly classified by the respective model. For each subset, the total number of the videos are between 7700 and 8000 and each class has at least 3, at most 20 instances. An adversarial version of the remaining 20% portion is generated with each adversarial attack, in a way that they cannot be correctly classified by the models. Then the adversarial set is used for evaluation along with the clean versions." Similarly to main experiments, we trained an MLP with 2 hidden layers with the same training hyper-parameters described in Supplementary F. For MVIT (Fan et al., 2021) and CSN (Tran et al., 2019) models, layers which have the name *conv_a* and *attn.proj* were filtered respectively. While for both of the models 3th 5th and 7th layers used to form subset $a_r$, for CSN (Tran et al., 2019) additional layers 4th, 6th and 8th were also used. $[:,: 4,: 200]$ and $[:,: 4,: 3,: 7,: 7]$ were used as slicing functions for MVIT (Fan et al., 2021) and CSN (Tran et al., 2019) respectively.

For Section 5.2, an MLP with 2 hidden layers with the same training hyper-parameters described in Supplementary F is trained. For Wav2vec (Schneider et al., 2019) model, layers which have the name *attention.dropout* were filtered, and 3th layer used to form $a_r$. As slicing function $[:,: 4,: 20,: 10]$ is selected. While 500 sound clips from LibriSpeech dataset used for experiments, 2000 sound clips saved for training.

For Supplementary H, we use the experimental settings as demonstrated in Mumcu & Yilmaz (2024b) and use the GTSRB (Stallkamp et al., 2012) dataset which includes 43 classes of traffic signs, split into 39,209 176 training images and 12,630 test images. LR detector is trained with the same settings that are specified for Resnet50 in Supplementary F.

## J    Visualized adversarial examples

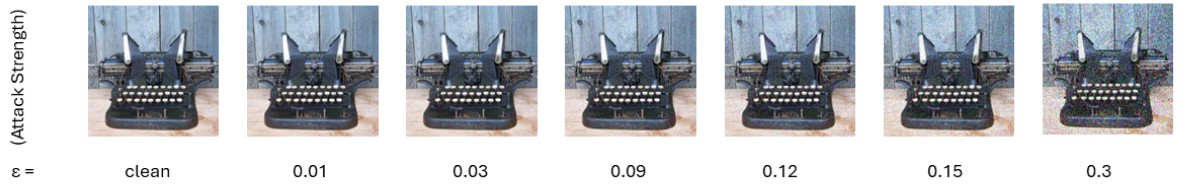

Figure 6: Effects of different attack strength values on a sample. The attack strength parameter $\epsilon$ is increased in the direction of the arrow. In the first sample, there is no attack. Samples are generated with adversarial attack PGD (Madry et al., 2017) and target model ResNet50 (He et al., 2015)
.

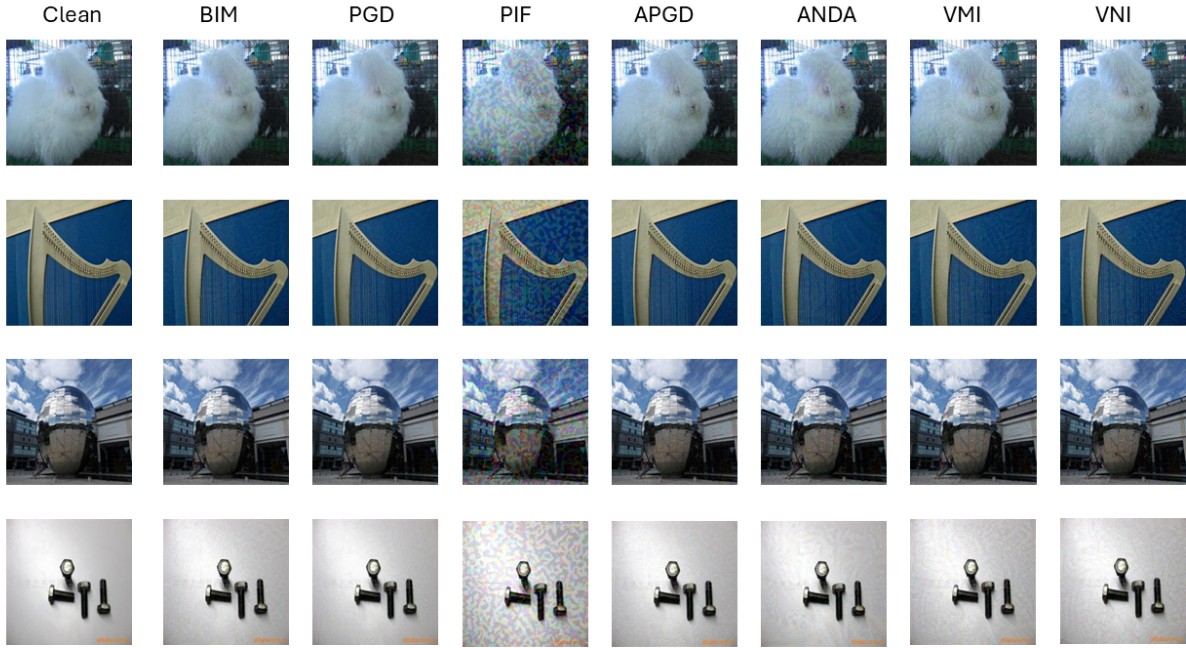

Figure 7: Adversarial examples generated with BIM (Kurakin et al., 2018), PGD (Madry et al., 2017), PIF (Gao et al., 2020), APGD (Croce & Hein, 2020b), ANDA (Fang et al., 2024), VMI and VNI (Wang & He, 2021) by attacking target model ResNet50 (He et al., 2015). First column represents the clean samples.

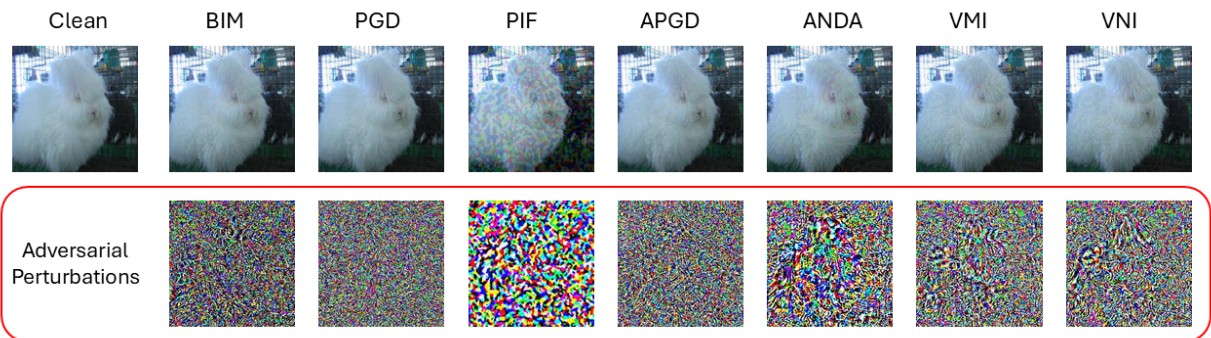

Figure 8: Visualization perturbations generated with BIM (Kurakin et al., 2018), PGD (Madry et al., 2017), PIF (Gao et al., 2020), APGD (Croce & Hein, 2020b), ANDA (Fang et al., 2024), VMI and VNI (Wang & He, 2021) by attacking target model ResNet50 (He et al., 2015). First row shows the clean sample and corresponding adversarial example for each attack. Second row demonstrates the noises added to the clean sample by each attack.

