# OpenReview forum: "Universal and Efficient Detection of Adversarial Data through Nonuniform Impact on Network Layers"
_TMLR — Accepted by TMLR_

### Review · Reviewer_jSMN · 2025-04-20

**Summary Of Contributions:**

The paper introduces a universal and efficient defense method against adversarial attacks. The proposed defense method is based on the conjecture that adversarial perturbations amplify as they propagate through the layers of a neural network.

Specifically, latent features extracted by multiple layers from the target model are fed into this MLP layer, and then the 'distance' between the output of this MLP layer and the feature vectors extracted by the target model is computed. If this  'distance' is larger than a predefined threshold, then the input is an adversarial sample.

**Audience:**

Yes

**Claims And Evidence:**

No

**Requested Changes:**

- In Eq.(3), the left-hand $g(x)$ denotes the vector of class probability (after softmax layer?), the right-hand $argmin_{w}$ returns the desired model parameters $w^\ast$, so it is inappropriate to say $g(x)=argmin_{w}$.
- In Eq.(4), what if $g(x)$ would not lead to the predicted label to be $y$? i.e., misclassification cases.
- For Eq.(5), adversarial perturbations amplify as they propagate through the layers of a neural network, which holds in most neural networks, but cannot be guaranteed. For example, if we have a 2-layer linear network $g(x)=W_2 W_1 x$ and $||W_2||<1, ||W_2||<1$. It would be better to be more rigorous.
- The rationale behind the selection of the mixture of early layer outputs (how to decide which layer should be selected) is not clear.
- Why not use KL divergence distance to measure the 'distance'?
- Why choose MLP with 2 hidden layers? What if the adversarial attacks consider to attack MLP as well?

**Strengths And Weaknesses:**

Strengths: The motivation is clear, and the perspective to detect adversarial examples via the conjecture that adversarial perturbations amplify as they propagate through the layers of a neural network seems interesting.

Weaknesses: 1. The writing needs to be significantly improved. 2. Lack of rigor, the conjecture (that adversarial perturbations amplify as they propagate through the layers of a neural network) holds in most neural networks, but cannot be guaranteed. 3. Lack of justifications for/rationale behind the proposed method's setting, like the number of MLP layers and the subset of layers for the MLP layers' input.

---

> ### Author Response · Authors · 2025-05-17
>
> - **The writing...**
>
> Thank you for your feedback. We have updated the paper to improve the writing and overall flow. This includes grammatical corrections, rewording, and rephrasing for clarity.
> - **Lack of rigor...**
>
> We agree that the conjecture is not guaranteed in all cases. However, this pattern is more related to the attack design than the target neural network and it has been observed in both prior work [1] and our empirical findings with static attacks. While we do not claim this behavior as a theoretical guarantee against any attack design, we use it as an empirical observation and motivation for the design of our universal and efficient detector. Our experiments demonstrate that this assumption holds sufficiently well across tested models and datasets in different domains (image, video, audio) to support effective detection against attacks that are ignorant to the detector.  Against adaptive attacks which know everything about our detector and are trained to deceive it, we observe that the detection performance drops to around 80% AUROC. This result is indeed an indicator that it is not easy to theoretically guarantee our underlying assumption that the first layer output for all attack designs remains close to the clean version. To clarify our underlying assumption, we now clearly state it in a new sentence added right before the Conjecture. It is now also emphasized with italic font in the Conjecture and explicitly stated in the Remark.
>
> [1] "Deep k-Nearest Neighbors:..." – Papernot and McDaniel, 2018
> - **Lack of justifications...**
>
> Regarding the number of MLP layers, we opted for a two-hidden-layer architecture to maintain a lightweight and efficient detector. Our goal was to keep the overhead minimal while still achieving high detection performance, and in practice, this architecture struck a good balance between complexity and accuracy. We added this explanation under LR Training in Section 4. As for the choice of input layers to the MLP, we analyze this in Appendix D by evaluating different combinations of early-layer outputs. We observed that including intermediate representations improves detection by capturing richer, yet relatively uncorrupted, features. We have also added a reference to this analysis in the main text for better visibility.
> - **In Eq.(3),...**
>
> We corrected Eq. (3) by rewriting it as $g(x) = g_{w^*}(x)$, where $w^* = \arg\min_w \mathcal{L}(g_w(x),y)$.
> - **In Eq.(4),...**
>
> We now explicitly state before Eq. (4) that we consider a highly accurate target model $g(x)$ that is trained to minimize the loss $\mathcal{L}(g_w(x),y)$. Since we do not aim to provide a formal proof, with this assumption, for our motivational purpose of developing a practical detector, we approximate the objective of maximizing the loss between $g(x^{adv})$ and $y$ with maximizing the loss between $g(x^{adv})$ and $g(x)$.
> - **For Eq.(5), ...**
>
> This assumption is more related to the attack design than the target neural network architecture. The provided example is missing the attack training objective of maximizing the loss for the targeted neural network. An effective attack must significantly modify the final layer output of the target model in order to cause an error. As stated in the response above to the “Lack of rigor” comment, as well as in the revised manuscript, we do not claim a theoretical guarantee for our assumption. We only use it in a conjecture to motivate the design of a practical lightweight and effective detector. With comprehensive experiments, we empirically justify our assumption and detector design against known attacks.
> - **The rationale...**
>
> We investigate the impact of early layer selection in Appendix D, where we compare different combinations and show how the choice affects performance. To improve clarity, we have added an explicit reference to this analysis in the main text.
> - **Why not use KL...**
>
> We interpret this comment as referring to the choice of distance metric used to compare the regressed feature vector to the actual feature vector from later layers. We opted for mean squared error (MSE) because the features being compared are real-valued vectors rather than probability distributions. KL divergence is appropriate when comparing probability distributions, such as softmax outputs. Since our regression operates in feature space, MSE is a more natural and stable choice in this context.
> - **Why choose MLP...**
>
> We chose an MLP with two hidden layers to keep the detector lightweight and efficient, as our goal is to develop a low-overhead detection method. This explanation is now added under LR Training in Section 4. In practice, this architecture performed well, and we found no need for deeper models. Additionally, we investigate the scenario where the attacker has full knowledge of the detection mechanism and targets both the main model and our MLP-based detector. This evaluation is presented in Section 4.4, Performance against Adaptive Attacks.

---

### Review · Reviewer_942J · 2025-04-24

**Summary Of Contributions:**

This paper introduces a method to detect Lp norm attacks against neural networks. The motivation for this method is that Lp norm attacks makes small perturbations to inputs but caused large changes to outputs. This led the authors to hypothesize that the magnitude of changes to a network activations increase across the network layers. Based on this, they introduce they are regression to identify these attacks using this heuristic. In tests, it works well against baselines.

**Audience:**

No

**Broader Impact Concerns:**

None.

**Claims And Evidence:**

Yes

**Requested Changes:**

I actually have very little to say here. I think the authors put things together very well. One thing that I would recommend would be to update the abstract to add a little bit more detail about how the method works. I just think that would be helpful.

**Strengths And Weaknesses:**

S1: I like this section on adaptive attacks. I think it's smart.

S2: Figure 3 is compelling.

S3: I think the paper is clear and well written. I like the way that it presents the conjecture before leading into the experiments.

S4: I really like section 5, showing that this works on other domains.

W1: this is a pretty minor concern. The paper I think the paper does a reasonable job of testing against baselines. However, in 2025, I admit I am a little bit surprised that stronger baselines don't exist or that something very similar to this hasn't already been done. In the related works section, I counted five mentions of papers that worked on prior detection method for Lp norm attacks. Consider me a little bit surprised if that's all there is.

W2: I think the biggest limitation of this paper is that it just focuses on Lp norm attacks. In the vision domain, these attacks are the most studied, but not necessarily the most hazardous type of attack against vision systems in the real world. Meanwhile, I think section 5 of the paper is really good. But overall, I think it's worth questioning the extent to which this paper is going to be very practically useful and interesting in 2025. I think it can be. But I think that the most compelling type of work on detecting adversarial attacks in 2025 is work that focuses on language models. Overall, I think that this paper does a good job and I think it's probably worth accepting. But someone who works in the field, I don't feel as if I have learned something very unexpected or important from reading this paper.

W2:

---

> ### Author Response · Authors · 2025-05-18
>
> - **I think the paper does a reasonable job of testing against baselines. However...**
>
> Thank you for this thoughtful observation. We appreciate your concern and would like to offer clarification on both the scope of our work and the current landscape of adversarial detection methods.
>
> First, we acknowledge that detection of Lp-norm adversarial examples has been studied in several prior works, some of which are cited in our Related Work section. However, many of these methods rely on assumptions such as access to adversarial examples during training, use of input transformations, or involve computationally expensive procedures like multiple forward passes or generative reconstruction. In contrast, our proposed Layer Regression (LR) method is post hoc, lightweight, model-agnostic, and does not require any changes to the original model or input. To the best of our knowledge, LR is the first approach to leverage early-layer features to regress later-layer outputs as a means to detect internal inconsistencies induced by adversarial perturbations, without the need for retraining, perturbing the input, or relying on external semantic priors.
>
> Additionally, we have updated the Related Work section in the revised version to include and briefly contrast two additional detection methods: Pang et al. (Reverse Cross-Entropy) and Tian et al. (Image Transformation Consistency), as suggested by another reviewer. This should help further to contextualize how LR differs in both methodology and applicability.
>
> - **I think the biggest limitation of this paper is that it just focuses on Lp norm attacks. In the vision domain...**
>
> Thank you for your thoughtful and constructive comments. We are pleased to hear that you found Section 5 strong and that you view the paper as a valuable contribution overall.
>
> Regarding your concern about the focus on Lp -norm attacks, we agree that these attacks do not cover the full spectrum of real-world threats. However, they remain the most widely adopted and systematically studied adversarial threat model in the vision domain, especially in white-box settings. From a defender’s perspective, attacks such as PGD and AutoAttack represent high-confidence, well-optimized adversarial examples that serve as essential benchmarks for evaluating any reliable detection method. We believe addressing this setting remains both relevant and necessary as a first step before expanding to more diverse or perceptual threat models.
>
> However, we agree that extending this direction to cover more realistic threat models, such as semantic or patch-based attacks, is both relevant and timely. Since LR is a general framework based on internal feature consistency, it can naturally be applied to these settings without relying on specific input assumptions.
>
> While our current work is centered around vision-based detection, we fully agree that adversarial robustness in large language models and multimodal systems is becoming increasingly important. We are actively exploring how the core principle behind LR, which is detecting inconsistencies between early and later layers, can be applied to these newer domains, including LLMs and VLMs. We see this as a natural and exciting extension of our approach.
>
> - **One thing that I would recommend would be to update the abstract...**
>
> Thank you for the feedback. As suggested, we updated the abstract to include a brief description of how the proposed method works. Specifically, we added a sentence summarizing the core mechanism behind our approach to improve clarity for the reader.

---

> > ### Comment · Reviewer_942J · 2025-05-21
> > **Thanks + response**
> >
> > > However, many of these methods rely on assumptions such as access to adversarial examples during training, use of input transformations, or involve computationally expensive procedures like multiple forward passes or generative reconstruction.
> >
> > This doesn't seem to apply to the field of anomaly detection. It's probably worth seeing what there is to add here.
> >
> > > We are actively exploring...
> >
> > This is a little bit vague. What's happening?

---

> > > ### Author Response · Authors · 2025-05-22
> > > **Response to "Thanks + response" by 942J**
> > >
> > > - **This doesn't seem to apply to the field of anomaly detection. It's probably worth seeing what there is to add here.**
> > >
> > > Thank you for the comment. There are indeed anomaly detection-based defense methods in the literature, such as Deep kNN, VLAD, and EPS-AD, which are discussed in the paper. These methods learn a baseline using clean (i.e., nominal) samples and compare the score of the test sample with the scores of the clean samples to detect any statistically significant deviation as an adversarial sample (i.e., anomaly). Unlike our proposed LR detector (which also follows an anomaly detection-based approach), the existing anomaly detection-based detectors are computationally expensive and not universally applicable to other domains like audio or video. Figure 3 shows that the proposed LR detector is orders of magnitude (more than 100 times compared to VLAD and more than 1000 times compared to EPS-AD) computationally more efficient than VLAD and EPS-AD. A similar comparison can be easily obtained with Deep kNN since it searches the k nearest neighbors in a high-dimensional space for each layer.
> > >
> > > - **This is a little bit vague. What's happening?**
> > >
> > > Thank you for the comment. To provide more clarity, we note that applying LR to LLMs and VLMs is not explored in this work but is part of our planned future work. While the core idea of detecting inconsistencies between early and later representations remains applicable, adapting it to these domains will require careful investigation of their architectural and modality-specific characteristics, which we envision for a follow-up study.

---

### Review · Reviewer_SVYA · 2025-05-13

**Summary Of Contributions:**

This paper addresses the challenge of identifying adversarial attacks in machine learning models. Specifically, it proposes a Layer Regression (LR) framework for detecting such examples. The LR method involves training a lightweight multi-layer perceptron (MLP) on clean data to estimate the output of each layer in a deep neural network (DNN). The authors demonstrate that adversarial samples have a disproportionate impact on the final layers of a DNN compared to earlier layers. By quantifying this difference, the LR approach can accurately identify adversarial examples. The authors then validate their method through extensive experiments and show its applicability to video action recognition and speech recognition tasks.

**Audience:**

Yes

**Claims And Evidence:**

Yes

**Requested Changes:**

I encourage the authors to address the first two weaknesses listed above. This should definitely strengthen this work further.

**Strengths And Weaknesses:**

Strengths:
* The proposed (LR) approach exhibits universality, demonstrated through various experiments.
*  The method offers reduced computational costs compared to existing approaches, making it more feasible for real-time applications.
* The observations that motivated the LR framework contribute to the novelty of this work.
* The experimental results are encouraging.


Weaknesses:
* While the paper presents a solid foundation, there is room for improvement in terms of presentation. Minor errors in grammar and consistency detract from the overall flow. I recommend revisions to address these issues.
* To further strengthen the paper's contribution, I suggest exploring connections to existing work in this domain, such as Towards Robust Detection of Adversarial Examples by Pang et. al and Detecting Adversarial Examples Through Image Transformation  by Tian, Yang and Cai.
* The authors demonstrate the LR method's versatility by applying it to video and speech recognition tasks. Considering its potential applicability to natural language processing (NLP) domains, particularly with the increasing adoption of large language models (LLMs), I encourage the authors to explore this area further.

---

> ### Author Response · Authors · 2025-05-18
>
> - **While the paper presents a solid foundation, there is room for improvement in terms of presentation...**
>
> Thank you for your feedback. We have updated the paper to improve the writing and overall flow. This includes grammar corrections, rewording, and rephrasing for clarity. All changes are marked in blue. Please see the revised version.
>
> - **To further strengthen the paper's contribution, I suggest exploring connections to existing work in this domain, such as...**
>
> Thank you for the suggestion. Towards Robust Detection of Adversarial Examples by Pang et al. proposes a training-time modification using reverse cross-entropy to enforce separable feature representations for clean and adversarial inputs, which improves detection performance but requires modifying the training process.
>
> Similarly, Detecting Adversarial Examples Through Image Transformation by Tian, Yang, and Cai introduces a detection method based on prediction consistency under image transformations, leveraging the instability of adversarial samples under rotation or translation. In contrast, our method operates on internal feature inconsistencies without requiring input perturbations or model retraining.
>
> This observation has also been incorporated into the revised version of the paper in the Related Work section, highlighted in blue.
>
> - **The authors demonstrate the LR method's versatility by applying it to video and speech recognition tasks. Considering its potential...**
>
> Thank you for your comment. We plan to extend our work in a future paper and explore LR's capabilities in multimodal systems, including LLMs and VLMs.

---

### Author Response · Authors · 2025-05-17
**Revision**

We thank all the reviewers for their thoughtful comments and suggestions. We have individually addressed all the concerns in our responses and in the revised manuscript. The changes in the manuscript are highlighted in blue.

---

### Decision · Action_Editor_WuX1 · 2025-06-10

**Recommendation:** Accept as is

**Audience:**

Yes

**Audience Explanation:**

There have been some (not completely unjustified) concerns regarding the novelty and impact of the work. I understand these concerns, and as I said above, I think the work adds another piece to a long line of work dating back to (at least) 2017. Nonetheless, the paper adds new knowledge that will be interesting to part of TMLR's audience (and significant novelty and impact are not required by TMLR's acceptance criteria).

**Claims And Evidence:**

Yes

**Claims Explanation:**

After the rebuttal and discussion phase, all reviewers argue that the claims made in the paper are supported by accurate, convincing and clear evidence, and I agree.

The paper proposes to train a lightweight detector for adversarial examples based on the observation that adversarial examples can often lead to substantial differences and OOD activation patterns in deeper layers, despite making minimal changes to the input and (potentially) very early layers. Reviewers appreciated the clear focus of the paper, and the practicality and reduced computational demand of the method, and think that results are convincing. Some reviewers point out that the paper only focuses on Lp-Norm attacks and that additional ablations could be interesting, and that the original submission needed improvements in terms of writing. The authors have addressed these concerns sufficiently in their responses.

I broadly agree with reviewers. The idea of using small detectors for adversarial inputs goes back at least to [On Detecting Adversarial Perturbations, Metzen et al. 2017](https://arxiv.org/abs/1702.04267), who also show that the defense network can easily be fooled by an attacker with gradient-access. Overall, I think the work presents some nice analysis and a solid incremental contribution. Taking all information together, I argue for accepting the work at TMLR.